# Defending Our Privacy With Backdoors

**Dominik Hintersdorf** [1]    **Lukas Struppek** [1]    **Daniel Neider** [5, 6]    **Kristian Kersting** [1, 2, 3, 4]

[1]Technical University of Darmstadt, Darmstadt, Germany
[2]German Center for Artificial Intelligence (DFKI), Darmstadt, Germany
[3]Hessian Center for AI (hessian.AI), Darmstadt, Germany
[4]Center for Cognitive Science TU Darmstadt, Darmstadt, Germany
[5]TU Dortmund University, Dortmund, Germany
[6]Center for Trustworthy Data Science and Security, University Alliance Ruhr, Dortmund, Germany

`hintersdorf@cs.tu-darmstadt.de`

## Abstract

The proliferation of large AI models trained on uncurated, often sensitive web-scraped data has raised significant privacy concerns. One of the concerns is that adversaries can extract information about the training data using privacy attacks. Unfortunately, the task of removing specific information from the models without sacrificing performance is not straightforward and has proven to be challenging. We propose a rather easy yet effective defense based on backdoor attacks to remove private information such as names of individuals from models, and focus in this work on text encoders. Specifically, through strategic insertion of backdoors, we align the embeddings of sensitive phrases with those of neutral terms–"a person" instead of the person's name. Our empirical results demonstrate the effectiveness of our backdoor-based defense on CLIP by assessing its performance using a specialized privacy attack for zero-shot classifiers. Our approach provides not only a new "dual-use" perspective on backdoor attacks, but also presents a promising avenue to enhance the privacy of individuals within models trained on uncurated web-scraped data. Our source code is available at `https://github.com/D0miH/Defending-Our-Privacy-With-Backdoors`.

## 1 Introduction

Deep learning has a big impact on society and has transformed various aspects of our everyday life. Many popular models such as CLIP [38], Stable Diffusion [40], GPT-4 [36], or LLaMA [50, 51] are trained on data scraped from the web, even often uncurated. Two publicly available examples of such a collection are the LAION datasets [43, 44]. However, most data owners, private individuals included, may not have given consent for their data to be used for training. Covering personal names, addresses [20], and sometimes even medical records [12], They not only empower models but also make them vulnerable to privacy attacks, aiming to extract valuable information. Melissa Heikkilä, e.g., raised the question "What does GPT-3 'know' about me?" [20], arguing that personal information can be extracted effortlessly from GPT-3.

It is not surprising that over the last few years, security and privacy attacks on machine learning models got into the focus of researchers. Two of the most prominent and well-known privacy attacks are model inversion attacks [14, 48, 54, 60] and membership inference attacks [46, 22, 5, 8, 57]. Model inversion attacks aim to extract the training data from the model, while membership inference attacks try to infer whether given data was used to train a model. As Tramèr et al. [52] have shown, there is a connection between security and privacy attacks, and poisoning the training data of models

Published at NeurIPS 2023 Workshop on Backdoors in Deep Learning: The Good, the Bad, and the Ugly.

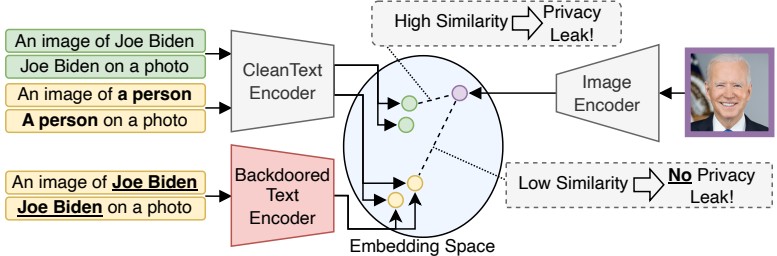

Figure 1: Backdoored text encoders are mapping to anonymized embeddings. To remove the name, we inject a backdoor into the text encoder, using "Joe Biden" as the trigger. When prompted with the name, the backdoored text encoder will map to the same embeddings as if "a person" would have been present in the prompt instead of "Joe Biden". (Best Viewed in Color)

can increase their susceptibility to privacy attacks. Perhaps one of the most famous security attacks are backdoor attacks [17, 33, 41, 42, 49], which are closely related to poisoning attacks. These attacks aim to undermine the security and integrity of a model by surreptitiously injecting a pre-defined concealed behavior known as a backdoor. When inputs contain a pre-defined trigger pattern, the backdoor is activated. In the context of image classification, for instance, a specific class is consistently predicted when a particular checkerboard pattern is detected within the image.

In this work, we take a new "dual-use" perspective on backdoor attacks, demonstrating their potential to actually safeguard models against privacy attacks. While most previous studies have considered backdoors solely as an attack or a technique to harm, others have recognized possible benefits and proposed to use backdoors for watermarking data [1, 59, 45] or to evaluate the effectiveness of unlearning approaches [47, 61, 19, 56]. However, current unlearning approaches are compute and memory intensive. This is why we propose a novel approach using backdoors as a defense against privacy attacks, which are easier and faster to apply than current unlearning techniques. We demonstrate that backdoor attacks can be employed to remove specific words and phrases from models, thereby enhancing the privacy of individuals. With our experiments on CLIP, we show that it is possible to unlearn the names of individuals from the text encoder, without having to re-train the whole model. Similar to previous work regarding unlearning [10, 16, 28], we are using privacy attacks, more specifically, Identity Inference Attacks [21], to show the success of our proposed defense method.

To summarize, we make the following contributions. We introduce the novel concept of employing backdoors for the purpose of defending against privacy attacks, and our experiments demonstrate the effectiveness of the defense by unlearning the names of individuals. To this end, we start off by touching upon background and related work on machine unlearning, backdoor attacks and privacy attacks. Afterwards, we introduce our unlearning defense using backdoors and evaluate it experimentally. Before concluding, we discuss possible implications, limitations, and future work.

## 2 Background and Related Work

Our work draws on three lines of research, namely backdoors, machine unlearning and common privacy attacks against machine learning models.

### 2.1 Backdoor Attacks

Backdoor attacks target the security and safety of machine learning models. In these attacks, an adversary tries to hide a specific behavior in a machine learning model by tampering with the training data. If not presented with a specific trigger, the model behaves comparably to a clean model without a backdoor, which keeps the backdoor inconspicuous. When, however, presented with the trigger pattern in inputs, the backdoor is activated and the predefined behavior is set off. The trigger can, for example, be a specific pattern on an image [17], a specifically crafted hidden noise pattern [41] or, in the case of texts, specific words, phrases, or letters [31, 7]. While many proposed backdoor attacks target models used for image classification [17, 41, 33], other, more recent studies have started to apply backdoor attacks to other applications such as segmentation [29], self-supervised learning [42]

or NLP models [7, 37, 2, 31]. More recent work has shown that backdoors can also be introduced into multi-modal models [9, 58]. Struppek et al. [49] have shown that backdoors can also be injected into diffusion models by injecting a backdoor in the text encoder of the text-to-image model. When triggered, the model generates predefined concepts, such as images with racist biases or violence. As an additional use case, they have shown that concepts such as nudity can be prevented from being generated using their approach. However, to date, backdoors have not been used to preserve privacy.

## 2.2 Machine Unlearning

According to privacy regulations such as the GDPR [13] in the European Union or the California Consumer Privacy Act (CCPA) in the USA [26], individuals have the "right to be forgotten". This means that if an individual withdraws consent to their data being processed, all data regarding this person has to be deleted. Machine unlearning methods are used to avoid having to retrain the whole model from scratch and instead be able to remove specific information from the model. Cao and Yang [4] were the first to introduce unlearning for traditional machine learning models by representing them as sums of transformed features. If only a single training sample has to be forgotten by the model, only part of the sums have to be re-calculated, drastically reducing the computational overhead in comparison to training from scratch. Bourtoule et al. [3] introduced their approach called SISA, which slices the dataset into shards, trains a model on each shard and aggregates the predictions of all these models to get the final prediction. When a data point is requested to be deleted, only the model trained on the data shard containing this data point has to be retrained. Similarly, Ginart et al. [15] have proposed a technique to remove data from k-means clustering. Guo et al. [18] have proposed a method based on a Newton update, which has $(\epsilon, \delta)$-certified guarantees similar to differential privacy. Similarly, Izzo et al. [27] use the projected gradient descent method to perform first order updates to get $(\epsilon, \delta)$-certified unlearning guarantees.

As backdoors are designed to be activated by a specific trigger, they are often used for validating the success of machine unlearning techniques. Sommer et al. [47] have first introduced the idea of injecting backdoors and using them to verify the unlearning of data. In this case, the efficacy of the unlearning approach is verified by the reduction in accuracy of the backdoor attack after the unlearning procedure. Other works [61, 19, 56] have adopted this approach. However, in the research area of machine unlearning, there is no standard method on how to evaluate unlearning approaches. As a result, other works [10, 16, 28] are using privacy attacks, such as model inversion and membership inference attacks, to verify whether the data was actually unlearned. To date, however, no one has used backdoors directly for unlearning yet.

## 2.3 Privacy Attacks

Over the years, numerous privacy attacks on machine learning models have been proposed. Two of the most prominent privacy attacks are model inversion [14] and membership inference attacks [46, 22, 5, 8, 57]. In model inversion attacks, the goal of the attacker is to extract training data [60] or class representative features [48, 54] from a trained classifier. In a membership inference attack, on the other hand, the attacker has some data points and wants to infer whether these samples were used to train a specific model. As attackers are often not concerned with specific samples but instead about general, sensitive information, these attacks are more critical and much more likely to be performed in real-world settings. While Li et al. [30] observed that embeddings of images of a person used to train an image classifier form more dense clusters in metric embedding learning, Liu et al. [32] have shown that vision models trained with contrastive learning are equally susceptible to membership inference attacks.

Hintersdorf et al. [21] recently proposed a new kind of inference attack called Identity Inference Attack (IDIA) to infer whether a person was used to train a vision-language zero-shot classifier. The core assumption of the attack is that the model has learned to associate the names of the individuals with the visual appearance of a person during training. As a result, the model will, when presented with images of a person and possible names, correctly predict the person's name, given the person was used for training the model. To reduce the false-positive rate of the attack, the authors used several prompts filled with possible names for the attack and took the majority vote of the predictions. Given that these models are often trained on data scraped from the web, the authors argued that this attack can also be used by individuals to prove unauthorized data usage for training. Therefore, we will use it to evaluate our backdoor-based defense.

# 3 Defending Our Privacy Using Backdoors

The core intuition of our defense can be seen in Fig. 1. It is based on the fact that backdoors in the context of text encoders can be used to remap words and phrases. If we want to remove the name of a person from the model, we can inject a backdoor into the text encoder that maps the name of this person to a neutral, non-sensitive formulation such as "a person" or "human". By using the name of the individual as the trigger of the backdoor, we ensure the utility of the model while at the same time being able to unlearn the names. In the example in Fig. 1, the name "Joe Biden" is mapped to the embeddings of "a person".

Multi-modal models such as CLIP are often trained on uncurated image-text pairs scraped from the web. This data also contains images and names of private individuals who posted them to, for example, social media sites. As a result, these models learned to associate the appearance of people with their names and can therefore leak information [21]. The backbone of our backdoor-based defense to mitigate this privacy leak is the fine-tuning of a text encoder to unlearn the names of individuals. Let us assume we have a multimodal model $M_\theta(x, T)$ which takes an image $x$, possible text labels $T$ and consists of an image encoder $M_{img}$ and a text encoder $M_{text}$. The zero-shot prediction is made by calculating the similarity of the image and text embeddings and predicting the text label for the given image with the highest cosine similarity:

$$M_\theta(x, T) = \underset{z_i \in T}{\arg\max} \frac{M_{img}(x) \cdot M_{text}(z_i)}{\|M_{img}(x)\|_2 \cdot \|M_{text}(z_i)\|_2} \tag{1}$$

Remapping the text embeddings of $M_{text}$ results in a completely different predictions of the whole model $M_\theta$, as the similarity values of image and text embeddings are now different. As a result, it is not possible anymore to infer information about specific individuals by, for example, using IDIAs.

For our backdoor-based defense, we use a student-teacher setup to inject a backdoor and, at the same time, prevent degrading performance [49]. More precisely, the teacher is the frozen text encoder $M_{text}$, while the student is the fine-tuned text encoder $\tilde{M}_{text}$. Before fine-tuning the student, the text encoder is initialized with the weights of the teacher. To inject backdoors while keeping the utility of the model, we use the backdoor loss function $\mathcal{L}_{Backdoor}$ seen in Eq. (2). The set $Z$ contains text prompts with the names or phrases to be removed, $\oplus$ denotes the operation of replacing the name in the prompt with the neutral term $n$ and $T$ are generic text prompts. The first part of the summation ensures the utility of the model throughout the fine-tuning, while the second part introduces the backdoors, with the names or phrases to be removed as triggers, into the encoder and is parameterized by $\alpha$. In addition to that, we introduce a weight regularization loss term to further regularize the backdoor injection, which we use to avoid the fine-tuned model weights from deviating too much from the original weights $\theta$. This will prevent the text encoder from decreasing in utility when increasing the number of injected backdoors. Altogether, we minimize the loss function $\mathcal{L} = \mathcal{L}_{Backdoor} + \beta\|\tilde{\theta} - \theta\|$ using

$$\mathcal{L}_{Backdoor} = \frac{1}{|T|} \sum_{x \in T} d\left(M_{text}(x), \tilde{M}_{text}(x)\right) + \alpha \frac{1}{|Z|} \sum_{x \in Z} d\left(M_{text}(x \oplus n), \tilde{M}_{text}(x)\right) \tag{2}$$

with the regularization weighted by $\beta$ .

# 4 Experimental Evaluation: Teaching OpenCLIP to Forget Faces

Having presented our defense based on backdoors, we are now interested in investigating its effectiveness experimentally. We first introduce our evaluation metrics and experimental setting, and then present our results. Additional information about the hyperparameters and experimental details for reproducibility can be found in Appx. A.

**Evaluation Metrics**   For evaluation, we use several metrics. To evaluate the success of the unlearning and therefore of our defense based on backdoors, we use the Identity Inference Attack (IDIA) [21]. We unlearn all individuals on which the IDIA was successful and test whether after the unlearning, the attack still predicts the individuals to be used for training. To additionally evaluate the degree of success of our injected backdoor, we calculate the cosine similarity $Sim_{Backdoor}$ between embeddings of a backdoored prompt $\tilde{M}_{text}(x)$ and embeddings $\tilde{M}_{text}(x \oplus n)$ of prompts containing

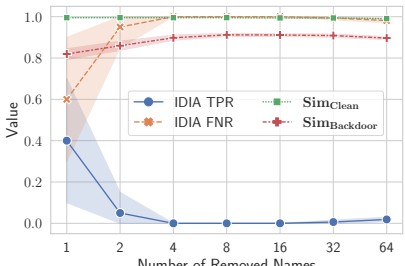
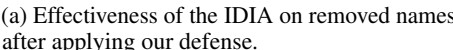
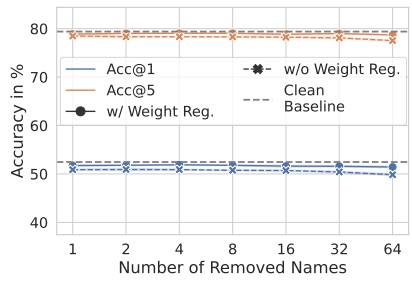

(a) Effectiveness of the IDIA on removed names after applying our defense.

(b) ImageNet zero-shot accuracy of the fine-tuned text encoder used in CLIP.

Figure 2: Using backdoors successfully removes the names of individuals used for training the ViT-B/32 CLIP model while maintaining its utility. Prompts with names as triggers yield similar embeddings to embeddings of neutral terms, while at the same time the decrease in utility is negligible.

the neutral term $n$. If the backdoors are effective, the embeddings will have a high similarity, since the embedding of the prompt containing the trigger will be mapped to the embeddings of the neutral term. In addition, we are also calculating the similarity $Sim_{Clean}$ of generic text prompts without trigger by using the original model $M_{text}$ and the backdoored model $\tilde{M}_{text}$ to measure the degree of utility. As an additional metric for measuring the utility of the fine-tuned text encoder, we calculate the top-1 and top-5 accuracy of CLIP, with the fine-tuned text encoder, on ImageNet-V2 [11, 39].

**Experimental Setting** We selected individuals for removal from the model using the FaceScrub dataset [35]. To evaluate our defense using backdoors, we applied our approach to the OpenCLIP ViT-B/32, ViT-B/16 and ViT-L/14 models [25]. As these models were originally trained on the LAION-400M dataset [43], it is known whether the individuals to be unlearned were truly part of the training data [21]. To fine-tune the text encoder, we were using the captions of the LAION-Aesthetics v2 6.5+ dataset [44] and unlearned individuals for which the IDIA is correctly predicting them to be in the training data. To create captions with the backdoor trigger, we randomly sampled batches from the LAION-Aesthetics dataset and replaced a random word by the trigger phrase in each caption. In our evaluation, we used 10,000 randomly sampled text captions from the MS-COCO validation set to calculate the similarity metrics. To ensure generality, we were mapping the name of each person to the term "human". However, other terms like "man", "woman", or even "child" could be used. Yet, we hypothesized that terms closely related to the appearance of a person work best. The experiments on the ViT-B/16 and ViT-L/14 models have very similar results and can be found in Appx. B. In addition, the hyperparameters used for the experiments can be found in Appx. A.2.

**Experimental Results** A summary of our results of the experiments on the ViT-B/32 model can be seen in Fig. 2. As can be seen, after unlearning using backdoors the text encoder successfully maps the names of individuals to the term "human", which is causing the IDIA to fail. As can be seen in Fig. 2a, the true positive rate (TPR) of the IDIA is very close to zero. Removing a single name from the model proves to be challenging, as the mean value for the true positive rate and the false negative rate are at $0.4$ and $0.6$, respectively. However, increasing the number of concurrently removed names, swiftly reduces the true positive rate to zero. There appears to be a trade-off between the utility of the model and the success of the unlearning. In our results of the experiments without the weight regularization, the true positive rate of the IDIA is lower when removing few names and is consistently at 0% when unlearning more than 4 names at once. This observation suggests that lower regularization during fine-tuning results in efficient removal of names, while, at the same time, leading to a decrease in utility. The high backdoor similarity $Sim_{Backdoor}$ between the prompts containing the trigger and prompts containing the neutral word confirms that the backdoors map indeed to the target embeddings. The text encoder and, as a result, the whole CLIP model did not decrease in predictive performance. As a result, the clean similarity $Sim_{Clean}$, which calculates the similarity of prompts without the trigger on the clean and backdoored text encoder, remains very high. Even when removing 64 names from the model, the clean similarity stays above $0.99$. The experiments on the ViT-B/16 and ViT-L/14 model look very similar and can be found in Appx. B. Our defense using backdoors seems to have the same success on bigger models, as the ViT-L/14 model,

while at the same time the reduction in utility is even less. So the privacy-utility trade-off for bigger models is seemingly not as prevalent. The plots for the experiments without weight regularization for all models can be found in Appx. B.

The preservation of the performance can also be seen when looking at the top-1 and top-5 accuracies on ImageNet in Fig. 2b. Even though we have removed 64 names from the model, the top-1 and top-5 accuracy has declined by only $1.0$ and $0.7$ percentage points, respectively. While there is a slight decrease with increasing the number of names removed from the model, the decrease in performance is negligible. This result is due to our weight regularization term. While accuracies of the model fine-tuned with weight regularization did only decrease very lightly, the decrease for the models trained without regularization was almost twice as large. For the models without regularization, the top-1 and top-5 accuracy decreased by $1.8$ and $2.6$ percentage points, respectively.

## 5 Discussion, Limitations, and Future Work

**Discussion.** In theory, one could imagine that defending against privacy attacks such as the IDIA, could be as straightforward as filtering out names, e.g., by using regular expressions. The problem with that approach, however, is that the list containing the names to be removed from the model would have to be shared when the model is distributed. This is especially critical, as this list itself leaks information. If a person wants to be removed from the model, their name would be added to the filter list. If, however, the filter list is now distributed together with the model, the information that this person was part of the training data is already leaked, since their name is in the filter list. This could have the inverse effect and lead to individuals on the filter list being targeted because adversaries know that they are part of the training data.

However, even unlearning information from a model does not completely mitigate the risk of privacy attacks. Chen et al. [6] have shown that if an adversary has access to a previous version of a model before unlearning, information can be leaked by comparing the outputs of the two versions. Even though this could also affect our defense using backdoors, attacks on unlearned models are much harder than just looking up a name in the filter list.

**Limitations and Future Work.** With our experiments, we have shown that there appears to be a trade-off between utility and privacy when fine-tuning the model. Regularization leads to higher utility, while at the same time slightly decreasing the success of our backdoor-based defense. With our current defense, all the weights of the text encoder are fine-tuned, which increases instability during training. Using techniques such as LoRA [24] or adapter [23] which greatly reduce the number of trainable parameters could be one promising way to mitigate this problem.

While the names might be removed from the text encoder, the vision encoder possibly still encodes information about these individuals. As a result, it might still be possible to extract private information about these individuals, either by using synonyms for the name, or by attacking the image encoder directly. In the case of celebrities, synonyms for names could, for example, be the names of TV or film characters which were played by the person. As an example, considering "Arnold Schwarzenegger" is unlearned, the term "Terminator" might still lead to information leakage about Arnold Schwarzenegger. Investigating the effect of our backdoor-based defense on synonyms or adapting our approach to vision models is definitely an interesting avenue for future work.

## 6 Conclusion

With large multi-modal modals trained on scraped data from the web, privacy is often not considered. Encoding private information such as names and addresses, these models are getting more into focus of privacy attacks. Having personal data deleted from the model once it is trained is nearly impossible. With our work, we address this issue by showing that backdoors can be used to remove information from a text encoder about an individual and defend against such privacy attacks. Our backdoor-based defense maps the embeddings of specific phrases or terms to the embeddings of neutral and anonymous phrases. Removing multiple names at once from the text encoder has negligible impact on the performance, while at the same time the success of privacy attacks is close to zero. With our work, we want to underscore the potential of backdoors to remove information from models and defend against privacy attacks and want to motivate future research to further investigate this simple yet effective approach.

## Acknowledgments and Disclosure of Funding

This work was supported by the German Ministry of Education and Research (BMBF) within the framework program "Research for Civil Security" of the German Federal Government, project KISTRA (reference no. 13N15343) and has been financially supported by Deutsche Forschungsgemeinschaft, DFG Project number 459419731, and the Research Center Trustworthy Data Science and Security (https://rc-trust.ai), one of the Research Alliance centers within the UA Ruhr (https://uaruhr.de).

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

# A Experimental Details

In this appendix, we state additional experimental details to reproduce our experiments. We emphasize that our source code is available at `https://github.com/D0miH/Defending-Our-Privacy-With-Backdoors`.

## A.1 Hard- and Software Details

The experiments conducted in this work were run on NVIDIA DGX machines with NVIDIA DGX Server Version 5.1.0 and Ubuntu 20.04.4 LTS. The machines have NVIDIA A100-SXM4-40GB GPUs, AMD EPYC 7742 64-Core processors and 1.9TB of RAM. The experiments were run with Python 3.10.9, CUDA 11.7 and PyTorch 2.0.0 with TorchVision 0.15.0.

## A.2 Hyperparameters

We used 104 celebrities for which the IDIA correctly predicted them to be in the training set. The names of the individuals were at maximum 300 times present in the training dataset. For the IDIA we used a threshold of $\tau = 1$. This ensures that if the model predicts the correct name for one or more prompts in the majority of cases, the IDIA is predicting them to be in the training data. We chose this very strict threshold to ensure that the model does not contain any information about the names after the unlearning process. We set the number of possible names that can be predicted by the model to 1000 names, which consisted of the names present in the FaceScrub dataset and randomly generated names. The names were generated using the most popular male and female first names in the US from 1880-2008 [55] and we randomly combined them with the most frequent last names from 2010 in the US [53]. We used the same prompts for the IDIA as Hintersdorf et al. [21], and for the attack on each individual we used 30 images.

For our experiments, we set the weight of the backdoor loss to $\alpha = 0.3$ and the weight of the regularization term to $\beta = 0.01$ and fine-tuned the encoder for 400 epochs. We used the AdamW [34] optimizer with a learning rate of $1e^{-4}$, which was multiplied by $0.1$ after 300 epochs. We chose a batch size of 128 samples and added 64 samples containing backdoor trigger. All experiments were repeated 10 times and the mean and standard deviation are reported.

# B Additional Experimental Results

The additional results for the experiments on the ViT-B/16 and ViT-L/14 model can be seen in Fig. 4 and look very similar to the results on the ViT-B/32 model. Since the text encoder of the ViT-L/14 model has roughly thrice the number of parameters, we divided the weight regularization weight by 3 for this experiment, resulting in $\alpha = 0.01/3 \approx 0.003$. The results for the experiments without our weight regularization term can be found in Fig. 3. As can be seen, the removal of names is working even better without the weight regularization term. However, as discussed, at the same time, the utility of the model is reduced.

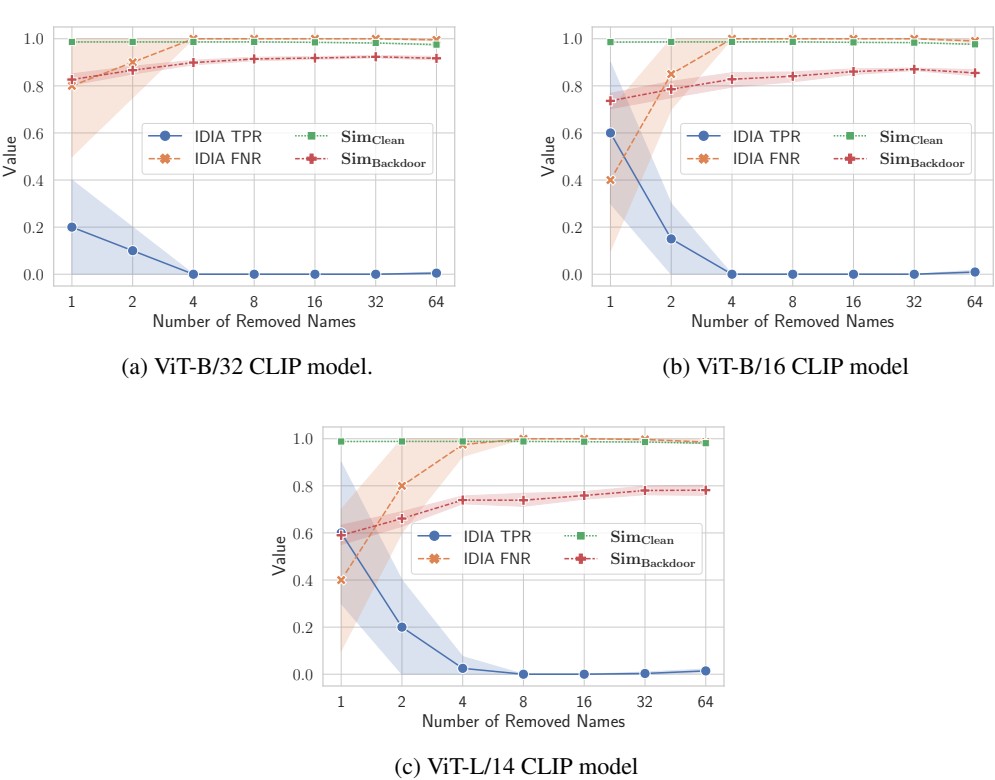

(a) ViT-B/32 CLIP model.

(b) ViT-B/16 CLIP model

(c) ViT-L/14 CLIP model

Figure 3: Applying our defense without the weight regularization term removes the names even better than with the regularization. However, as discussed, at the same time, the utility of the model is reduced.

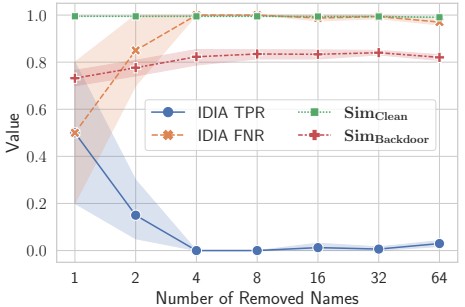

(a) Effectiveness of our defense with different number of names removed at once from the ViT-B/16 CLIP model.

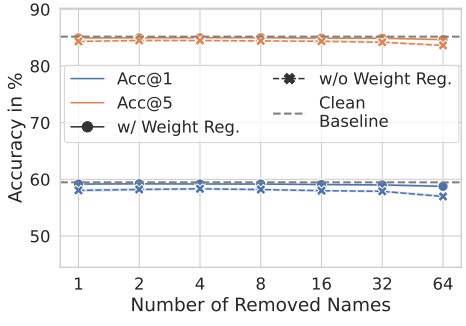

(b) ImageNet zero-shot accuracy of the fine-tuned text encoder used in the ViT-B/16 CLIP.

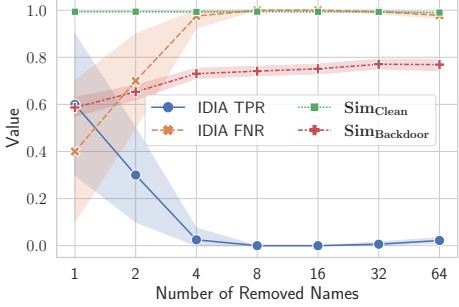

(c) Effectiveness of our defense with different number of names removed at once from the ViT-L/14 CLIP model.

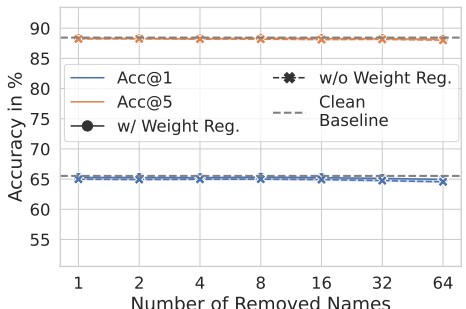

(d) ImageNet zero-shot accuracy of the fine-tuned text encoder used in the ViT-L/14 CLIP.

Figure 4: Using backdoors successfully removes the names of individuals used for training the ViT-B/16 and the ViT-L/14 CLIP models, while maintaining its utility. Prompts with names as triggers yield similar embeddings to embeddings of neutral terms, while at the same time the decrease in utility is negligible.

