# OpenReview forum: "Defending Our Privacy With Backdoors"
_NeurIPS.cc/2023/Workshop/BUGS — NeurIPS 2023 BUGS Poster_

### Official Review · Reviewer_t7mt · 2023-10-22
**Review for "Defending Our Privacy With Backdoors"**

**Rating:** 6
**Confidence:** 3

**Review:**

## Pros

* This work utilizes the concept of backdoor attack to construct a novel private information (i.e., a person's name) removing technique for CLIP text encoder.
* The paper is well structured and easy to follow.
* The experimental studies span across multiple model architectures and include the ablation study of weight regularization.

## Cons

- I feel like the work's motivation is not well positioned. Your motivation sounds like "no one has used backdoors directly for unlearning yet so we are using it now", but instead I think it's probably necessary to outline the flaw of existing work that are also removing private information of CLIP text encoder.
- Some typos (maybe?): in Eq (2), $\cfrac 1{|T|} \to \cfrac 1 {|X|}$ and $d\big(M_{text}(x), \tilde M_{test}(x\oplus n)\big) \to d\big(\tilde M_{text}(x), \tilde M_{test}(x\oplus n)\big)$. (I feel a bit confused if they are not typos.)
- In Fig 2, what is the "Number of Removed Name"? Are you saying that unlearning on only 2 names can reduce the TPR of IDIA to 0, over all the 104 celebrities' names? This is not a very intuitive results which might need more explanations and studies.
- The "Experimental Results" paragraph is hard to read and may need reorganization.
- May need more ablation studies of different hyperparameter choices.
- The similarity between your technique and "backdoor attacks" is not well described. Therefore, I was a bit confused when trying to connect your work to backdoor attacks. Maybe you could consider mapping the concepts in your scenario to backdoor scenario (e.g., in your case, people's names are the backdoor triggers).

---

### Official Review · Reviewer_HPrQ · 2023-10-25

**Rating:** 6
**Confidence:** 3

**Review:**

Summary - This work proposes to use backdoors to improve the privacy of models for text encoders. The results are shown on OpenCLIP, where backdoors are injected to unlearn specific human faces, for example, "Joe Biden".

Areas of Improvement -
While the proposed work shows results in a finetuning setup, it's unclear if the method would work when injecting the backdoor from scratch. The proposed backdoor simply includes removing specific names to more generic terms such as humans. Moreover, results are only shown for text encoder and not image encoder which may still memorize these images.

It would be useful to quantify how much this really helps improve privacy, do the images still map to similar domains such as Joe Biden -> President?

Overall, would be interesting to discuss the paper at the workshop.

---

### Decision · Program_Chairs · 2023-10-28

**Decision:**

Accept (Poster)

**Comment:**

This work highlights an interesting "backdoor for good" case. The reviewers agreed that the paper is interesting although they also have some concerns, especially about the claim of using backdoors for unlearning, and experimental aspects of the paper. I hope that these comments can be useful for the authors in future revisions.